# Mixed Depression: A Mini-Review to Guide Clinical Practice and Future Research Developments

**DOI:** 10.3390/brainsci12010092

**Published:** 2022-01-11

**Authors:** Antimo Natale, Ludovico Mineo, Laura Fusar-Poli, Andrea Aguglia, Alessandro Rodolico, Massimo Tusconi, Andrea Amerio, Gianluca Serafini, Mario Amore, Eugenio Aguglia

**Affiliations:** 1Psychiatry Unit, Department of Clinical and Experimental Medicine, University of Catania, 95123 Catania, Italy; ludwig.mineo@gmail.com (L.M.); laura.fusarpoli@gmail.com (L.F.-P.); alessandro.rodolico@phd.unict.it (A.R.); eugenio.aguglia@unict.it (E.A.); 2Section of Psychiatry, Department of Neuroscience, Rehabilitation, Ophthalmology, Genetics, Maternal and Child Health, University of Genoa, 16126 Genoa, Italy; andrea.aguglia@unige.it (A.A.); andrea.amerio@unige.it (A.A.); gianluca.serafini@unige.it (G.S.); mario.amore@unige.it (M.A.); 3IRCCS Ospedale Policlinico San Martino, 16132 Genoa, Italy; 4Section of Psychiatry, Department of Medical Sciences and Public Health, University of Cagliari, 09042 Cagliari, Italy; massimotusconi@yahoo.com

**Keywords:** mixed states, mixed depression, mood, bipolar disorder, psychomotor agitation, review, Koukopoulos

## Abstract

The debate on mixed states (MS) has been intense for decades. However, several points remain controversial from a nosographic, diagnostic, and therapeutic point of view. The different perspectives that have emerged over the years have turned into a large, but heterogeneous, literature body. The present review aims to summarize the evidence on MS, with a particular focus on mixed depression (MxD), in order to provide a guide for clinicians and encourage the development of future research on the topic. First, we review the history of MS, focusing on their different interpretations and categorizations over the centuries. In this section, we also report alternative models to traditional nosography. Second, we describe the main clinical features of MxD and list the most reliable assessment tools. Finally, we summarize the recommendations provided by the main international guidelines for the treatment of MxD. Our review highlights that the different conceptualizations of MS and MxD, the variability of clinical pictures, and the heterogeneous response to pharmacological treatment make MxD a real challenge for clinicians. Further studies are needed to better characterize the phenotypes of patients with MxD to help clinicians in the management of this delicate condition.

## 1. Introduction

For centuries, there has been a continuous debate on mixed states (MS). Nevertheless, the co-existence of elements belonging to opposite polarities in the same clinical picture, the variability of manifestations and complications, as well as the peculiar pharmacological treatment, still make MS a challenge for clinicians. Several points remain controversial. First, it is difficult to systemize MS within the common nosographic systems, as they could be considered either as independent entities or be collocated along a spectrum ranging from pure depression to pure mania. Second, it is unclear which symptoms need to be encompassed in the clinical picture and which scales and diagnostic criteria should be adopted in clinical practice for a prompt identification of MS. Third, there is scant evidence of the most effective pharmacological treatment in both acute-phase management and maintenance. Notably, the most up-to-date guidelines have difficulty in systematizing these entities and providing precise indications for clinicians [1]. For these reasons, the present review aims to comprehensively summarize the evidence on MS, with a particular focus on mixed depression (MxD), in order to provide a guide for clinicians and to encourage the development of research studies in the future.

## 2. Brief Historical Overview of the Concept

The concept of manic–depressive illness, including MS, was originally formulated in classical medicine, starting with Hippocrates (ca. 460–370 B.C.) and Aretaeus of Cappadocia (ca. 30–130 A.C.) [2,3]. The earliest evidence of an established MS can be found in ancient medical manuals as well as in 18th-century treatises on psychopathology, although the first conceptual and terminological definitions only date back to the 19th century [4,5].

In the field of psychiatry, among the pioneers of the exploration of MS, Heinroth—in his work entitled “Disorders of Mental Life or Mental Disorders”—used the German term “*Mischungen*” (“mix or mixture”) to identify psychopathological conditions difficult to classify because of the coexistence of contrasting affective elements [6]. Wilhelm Griesinger, another German psychiatrist, described states of mental alteration characterized by the coexistence of melancholic and manic elements and distinguishable from rapid cycling forms and seasonal affective disorders. Griesinger defined those psychopathological conditions as “intermediate forms,”, i.e., “melancholia with destructive impulses” and “melancholia with long-lasting exaltations of volition” [7].

Despite the contributions of previous psychiatrists in describing clinical pictures assimilated to the current concept of MS, Emil Kraepelin was the first author to systematize MS, including these conditions within the context of manic–depressive illness [8]. Indeed, Kraepelin and his pupil Weygandt conceived depressive and manic phases as the utmost of a unique disorder, namely manic–depressive illness [9], and considered MS as the most common forms of this disorder [10]. On the contrary, in the 1950s, Karl Leonhard posited a clear distinction between unipolar and bipolar disorder [11]. Modern nosography is fundamentally based on Leonhard’s dichotomy [12]. Consequently, MS have only been studied marginally for decades.

## 3. Mixed States in the DSM: From the Revolution of DSM-III to DSM-IV Mixed Episode

In the first edition of the DSM (1952), MS were confined in a marginal position as part of the category “manic depressive reaction, other” [13]. The DSM-II (1968) introduced the category of “mixed manic-depressive illness”, collocating it among “Other major affective disorders”, describing it as a condition in which “manic and depressive symptoms appear almost simultaneously” [14].

In 1980, the DSM-III radically changed the classification of mood disorders, as Kraepelin’s broad concept of manic–depressive insanity was separated into two distinct entities: bipolar disorder (BD) and major depressive disorder (MDD) [15]. This classification was adopted after Leonhard’s dichotomy and did not take into account Kraepelin’s conception of manic–depressive illness as a continuum [16]. Additionally, the DSM-III introduced the diagnostic category of “Bipolar disorder, mixed”, defined as “current (or most recent) episode involving the full symptomatic picture of both manic and major depressive episodes, intermixed or rapidly alternating every few days.” This definition was substantially retained in the DSM-III-R (1987), except for the duration criterion of two weeks for depressive symptoms [17]. In the DSM-IV and DSM-IV-TR, MS were incorporated into the diagnostic category of “mixed episode” [18,19]. According to DSM-IV-TR criteria, it was possible to diagnose a mixed episode in the co-presence of criteria to diagnose either a manic or major depressive episode (except for the time criterion) for at least one week. Therefore, a mixed episode could only be present in the case of a BD-I diagnosis and not in MDD or BD-II. In fact, manic episodes are pathognomonic for BD-I but not BD-II, which is instead characterized by an alternation of hypomanic and depressive episodes. The DSM-IV-TR operational definition of a mixed episode has proved to be extremely narrow, as it targeted an almost unrealistic clinical condition, failing to discriminate the most prevalent presentations of MS, i.e., sub-threshold forms characterized by the occurrence of few symptoms of the opposite polarity during the same affective episode [20,21].

## 4. Current Nosographic Classification of Mixed Episodes and Mixed Depression

The precise identification of a depressive episode with mixed characteristics is of great importance for the subsequent diagnostic framework and treatment planning because of the significantly worse course and the peculiar responsiveness to treatment [22]. The DSM-5 has represented, from this point of view, a significant step forward, with the introduction of the specifier “with mixed characteristics”; for the first time, this clarification has enabled the recognition of the possible co-presence of expansive and depressive symptoms during a depressive episode within a BD (Type I or Type II), or MDD [23]. Indeed, in the DSM-5, a depressive episode with mixed features can be diagnosed when “full criteria are met for a major depressive episode, and at least three manic/hypomanic symptoms are present during the majority of days of the current or most recent episode of depression”. Of note, distractibility is not enumerated among manic/hypomanic symptoms that can appear during a mixed episode [24]. There is growing consensus that the introduction of the specifier of “mixed characteristics” may allow clinicians to describe mood disorders along a spectrum ranging from pure unipolar depression to pure mania [25].

Differently from the DSM-5, the 11th edition of the International Classification of Diseases (ICD-11), classifies MS as a separate diagnostic category. MS are defined as “the presence of several prominent manic and depressive symptoms consistent with those observed in manic episodes and depressive episodes, which either occur simultaneously or alternate very rapidly…” [26]. It is specified that, when depressive symptoms predominate in a mixed episode, the most common contrapolar symptoms are irritability, distractibility, increased verbal production, and psychomotor agitation [27]. Therefore, unlike the DSM-5, the choice of maintaining a specific diagnostic category and including the overlapping symptoms may increase diagnostic sensitivity and ensure more targeted therapeutic strategies. Clinicians should also be aware that the ICD-11 introduced the diagnostic category “mixed depressive and anxiety disorder”, which may represent a potential confounding factor. This category is defined as the co-presence of depressive and anxiety symptoms, not severe enough to warrant a diagnosis of depressive disorder or anxiety disorder [26]. Therefore, this diagnostic category does not deal with MS or MxD.

Some authors criticize the choice of both current nosographic models to propose a definition of MS that involves a combinatorial approach of symptoms of both polarities rather than specific profiles of different mixed episode subtypes. Indeed, the combinatorial approach would not be able to discriminate among the more serious clinical forms of MxD [28]. Another consideration must be made about the choice of including also the rapid alternation of the two polarities in the definition of MxD. This makes it difficult to discriminate between a mixed episode and ultra-fast cycling (mood-changing states of polarity over weeks or days) and, in particular, ultradian cycling (mood changes occurring within a day) [29].

## 5. The Concept of Mixed Depression and Alternative Models to Official Nosographic Categories

Some authors disagree with the conceptualization of mixed depression (MxD) proposed by the DSM-5. The most relevant aspect is related to the set of diagnostic criteria proposed for the definition of MxD, which, according to some authors, would be too narrow to include the entire spectrum of symptoms. One criticism is the absence of irritable mood and psychomotor agitation in the current DSM-5 diagnostic category of “depressive episode with mixed features” [30]. This is important because prevalence rates may vary depending on the definition of MxD used. For instance, a study by Miller et al. (2016) found that the use of narrower criteria (i.e., DSM-5) allowed them to identify 2.6% to 10.8% of MxD in a sample of patients with BD. Conversely, 14.9% could be diagnosed with MxD using broader criteria (i.e., the presence of subthreshold hypomania concurrent with at least mild depression) [31].

Koukopoulos’ studies are among the proposed alternative models. In the early 1990s, Koukopoulos described a group of patients with apparently classical depression who did not respond to treatments with antidepressants. This group of patients manifested psychomotor agitation, insomnia, sometimes increased suicidal ideation, and psychotic symptoms. The main characteristics described were intense emotionality and the marked expression of feelings with crying crises and a cyclothymic or hyperthymic temperament [32]. Koukopoulos hypothesized that this presentation could originate from an excitatory process, as justified by the response to treatment with antiepileptics, neuroleptics, lithium, and electroconvulsive therapy [33]. In 2007, Koukopoulos proposed new diagnostic criteria for the classification of MxD, including psychomotor agitation, irritability, and mood lability [34]. This classification was further validated by a larger clinical study [12,35]. In empirical studies, the frequency of mixed-mood states similar to the DSM-5 definition ranges from 7 to 12% [36]. In contrast, the inclusion of irritability and psychic or psychomotor agitation as central features of MxD increased the frequency of mixed mood states up to 47% [34]. For this reason, Koukopoulos and Sani suggested the possibility of renaming the clinical identity of these symptoms as “excitatory” instead of “manic”, highlighting a different nature of this component than the pure manic symptom [34]. Of note, a recent multicenter study showed an overlapping discriminant capacity between traditional DSM-5 criteria and Koukopoulos’ mixed-depression criteria [37]. Moreover, the cross-sectional, multinational study “Bipolar Disorders: Improving Diagnosis, Guidance and Education (BRIDGE)-II-MIX” called for the inclusion of symptoms such as psychomotor agitation, mood lability, hypersexuality, and aggressiveness in the DSM-5 specified for “mixed episodes” [23].

Benazzi, another Italian author that focused his research on MS, considered MxD within the dimensional approach of mood disorders. In contrast with Koukopoulos, the definition of MxD proposed by Benazzi refers to a minimum number of hypomanic symptoms (without specifying which one in particular) present within the depressive state [38]. According to the author, this definition would increase the diagnostic sensitivity of MxD [39]. The cut-off for a diagnosis of MxD is the presence of three symptoms of hypomania or a score >= 8 in the Hypomania Interview Guide (HIG) during a depressive episode [40].

More recently, Mahli and colleagues proposed the Activity, Cognition, and Emotion (ACE) model as a possible approach for studying mood disorders. This model divides the different symptoms within the three dimensions, considering how they may vary over time. This model is in line with Kraepelin’s conceptualization of manic–depressive illness as a spectrum [41]. Any symptoms within the domain can be considered as primary or secondary or described by a scale of severity from non-pathological to characterizing. This model allows a greater understanding of the pathophysiology underlying the disorder. In addition, it allows the conceptualization of the various subclinical aspects of clinical presentations by improving the recognition and understanding of mood disorders [41,42,43,44,45].

Unfortunately, at present, although the DSM-5 with the mixed specifier may add a feature to unipolar MxD, there is speculation that the presence of overlapping excitatory symptoms may more accurately describe a subgroup of patients with MDD with marked bipolar predisposition [22]. These considerations suggest the need for a significant revision of the specifier regarding depression with mixed features in the next revision of the DSM-5 [46].

## 6. Mixed Depression in Clinical Practice: Diagnosis and Course of Illness

The clinical characteristics of MxD are widely variable. Typically, depressive symptoms of mixed and non-mixed episodes substantially overlap in their clinical presentation and severity [47]. Negative self-evaluation, increased energy, and racing thoughts seem to be present in MxD [48]. High levels of anxiety characterize the clinical presentation of MxD, although its diagnostic usefulness is limited, because anxiety is also predominant in non-mixed depressive states [2]. The same applies to agitation, understood both as internal tension and motor restlessness. Indeed, several studies have shown the presence of agitation even in the case of non-MxD [49]. Psychotic symptoms may also occur in the clinical context of MxD [50]. In fact, patients who present mixed symptoms differ from those with pure forms by higher suicidality rates, higher relapse rates, the higher incidence of comorbidities, and lower rates of response to treatment.

Some works in the literature have reported that patients with MxD are more frequently females, show higher rates of medical and psychiatric comorbidities and fewer euthymic mood periods [1,31]. The pediatric population represent a poorly studied category which, however, seems at risk for the development of mixed symptoms. In fact, adolescents with MxD appear to have higher levels of disability, increased severity, and more comorbidities than adolescents with pure MDD or BD [51]. During depression, even a family history of BD or completed suicide could be indicative of the possible development of mixed symptoms and manic switch [52]. Finally, taking antidepressants during a depressive episode may represent an important cause of development of mixed symptoms (i.e., iatrogenic cause). In this case, the main guidelines recommend a careful reduction in the dosage or even a suspension of the antidepressant medication [1].

The combination of despair experiences with increased energy and impulsivity makes MxD patients more at risk for suicidal behavior [53]. High levels of anxiety can further increase the risk of suicide [54]. As reported by the DSM-5, patients with mixed features have to be strictly monitored and followed-up, as the presence of subthreshold manic symptoms represents a phenotypic indicator of a bipolar diathesis. Therefore, the presence of hypomanic symptoms may indicate a risk factor for a subsequent development of a BD [55].

Regardless of the risk of a possible transition to a full-blown BD, the presence of mixed features in the context of a major depressive episode is associated with a more severe illness phenotype, with more likely mixed relapses.

## 7. Psychometric Tools to Evaluate Mixed Depression

Several scales have been identified to assess the presence and severity of (hypo)manic symptoms within the clinical presentation of psychiatric patients, such as the Young Mania Rating Scale (YMRS; [56]), the Internal State Scale (ISS; [57]), the Hypomania Checklist-32 (HCL-32; [58]), and the Altman Self-Rating Mania Scale (ASRM; [59]. Other scales, such as the Montgomery–Åsberg Depression Rating Scale (MADRS) or the Hamiton Depression Rating Scale (HAM-D) can be used to assess the presence and severity of depressive symptoms. The combination of scales for depression and mania can be useful for clinicians to assess the presence of MS even if they are not specifically designed for this purpose.

Interestingly, the Bipolar Disorder Rating Scale (BDRS, [60]) is an instrument specifically designed for bipolar depression. It is a semi-structured interview composed of 20 items, each with a score that can range from 0 to 3 for a maximum total of 60 points.

Indeed, only a few scales have been specifically created to identify (hypo)manic symptoms in the context of MxD. The Mini-International Neuropsychiatric Interview (MINI) consists of several modules that allow the identification of the main psychiatric diagnoses, with dichotomous items (yes/no). Recently, in the section that refers to depressive disorder, a subscale for MxD has been formulated [61]. This scale is self-rated by the patients.

The Clinically Useful Depression Outcome Scale with questions for DSM-5 Mixed subtype (CUDOS-M, [62]). It is a self-assessment scale developed by Zimmerman et al. (2014), which includes 13 items based on the DSM-5 specific criteria for MxD with satisfactory reliability and validity [63]. Each item of the scale is rated on a 5-point ordinal scale to indicate the frequency of symptoms during the past week (0 = not at all true [0 days]; 4 = almost always true [every day]), and the total score ranges from 0 to 52.

The Structured Clinical Interview for DSM (SCID) is a semi-structured interview organized into modules for formulating DSM-5 diagnoses [64]. In the latest version, in the section corresponding to depressive disorders, the symptomatology of MxD is also investigated.

The Koukopoulos Mixed Depression Rating Scale (KMDRS) was developed by Koukopoulos to evaluate the symptoms of MxD based on the criteria he proposed (i.e., the presence of a depressive episode plus irritability and internal tension, emotional lability, and absence of slowing down). The KMDRS is a self-administered scale, consisting of 14 items evaluating the presence and severity of the excitatory symptoms typical of MxD [65]. Moreover, Tavormina et al. recently developed the G.T. Mixed States Rating Scale (GTMSRS), a self-administered rating scale composed of 11 items, of which seven also include sub-items [66,67].

Finally, the Clinical Monitoring Form (CMF) proposed by Sachs et al. may represent a useful, freely available tool in high-risk cases, such as patients with depressive symptoms and a family history of BD [68].

## 8. Pharmacotherapy

The pharmacological management of MS has always represented a challenge for clinicians who must balance the need to treat both manic and depressive symptoms with the risk of mood-switching. In fact, the use of antidepressant medications to treat depressive symptoms can induce a switch to mania [69]; conversely, a pharmacotherapy based on antipsychotics (especially in the case of strong D2 receptor blockers) may increase the risk of switching to depression [70,71].

A further difficulty is that the available literature evidence is scant and weakened by important methodological limitations that affect randomized clinical trials. The evaluation of the response to medications of (hypo)manic patients with depressive symptoms is mainly based on the post-hoc or pooled analysis of randomized clinical studies originally meant to study treatment efficacy in manic episodes. Since the presence of contrapolar symptoms generally constitutes an exclusion criterion in trials conducted on subjects affected by major depressive episodes, the evidence for MxD is even more lacking. Hence, historically, pharmacotherapy of MS has represented an unmet need in the international guidelines for the treatment of mood disorders.

To date, only three international guidelines specifically address the treatment of MS: the Canadian Network for Mood and Anxiety Treatments (CANMAT) and the International Society for Bipolar Disorders (ISBD) recommendations (2021, [1]); the World Federation of Societies of Biological Psychiatry (WFSBP) guidelines (2019; [72]; the guidelines developed by Stahl and colleagues focused exclusively on MxD (2017; [73]). Moreover, treatment recommendations for episodes with mixed features are available in the updated editions of some international guidelines for BD: Evidence-based guidelines for treating bipolar disorder: revised third edition recommendations from the British Association for Psychopharmacology (2016; [74]); the International College of Neuro-Psychopharmacology (CINP) treatment guidelines for bipolar disorder in adults (2017; [75]); Royal Australian and New Zealand College of Psychiatrists (RANZCP) clinical practice guidelines for mood disorders (2021, [76]). The guidelines adopt different definitions of MS or MxD, which are displayed in Appendix A.

Despite their heterogeneity, all the guidelines agree upon avoiding the use of antidepressants in MxD in monotherapy. In cases in which an antidepressant is needed, they suggest combining it with a mood stabilizer or a second-generation antipsychotic (SGA). Of note, only Stahl’s guidelines expressly include the use of antidepressants in combination with mood stabilizers or SGAs among the third-level recommendations (lithium or lamotrigine or valproate or atypical antipsychotic + bupropion; lithium or lamotrigine or valproate or atypical antipsychotic + selective serotonin reuptake inhibitors; lithium or lamotrigine or valproate or atypical antipsychotic + monoamine oxidase inhibitors).

SGAs are psychotropic agents that are generally considered as first-line or second-line choices in the treatment of acute depression with mixed features by most of the guidelines considered (Table 1). Among antipsychotics, olanzapine, lurasidone, and ziprasidone appear to have more evidence of efficacy in treating MxD [69]. The use of mood stabilizers, such as lithium or valproate, is recommended during the maintenance phase, although they can also be prescribed during the acute phase [73,76]. Finally, the usefulness of lithium in preventing suicide is evident: data show its efficacy even when it is not effective in preventing affective episodes [77].

## 9. Conclusions

Our study aimed to provide a synthesis about MxD and guide clinicians’ choices. The present review shows that current evidence is still unclear in several points. This uncertainty may be related to the lack of neurobiological and epidemiological studies based on recognized diagnostic criteria [78]. A better phenotyping of patients in clinical practice could solve many questions about the diagnostic orientation [79]. In this regard, the creation of clinical and research groups regarding MxD would favor the formulation of more sensitive and specific criteria for the identification of this condition. This may consequently lead to a personalization of care and treatment. In fact, the use of inappropriate medications, such as antidepressants, may worsen the clinical picture instead of ameliorating symptoms. Of note, longitudinal studies are warranted to evaluate whether the presence of mixed features in individuals with MDD may represent a warning for a potential evolution to BD. The present review shows that it is important for clinicians to remember that MxD is very common in clinical practice and is often associated with a worse outcome and high suicide risk. Future research should focus on these aspects to provide clear answers.

## Figures and Tables

**Table 1 brainsci-12-00092-t001:** Pharmacological treatment for the acute phase of mixed depression (MxD).

	CANMAT—ISBD: DSM-5		CANMAT—ISBD: DSM IV		WFSBP		CINP		RANZCP *	Stahl
	D	M	D	M	M	D	M	D		
**First-line**			Asenapine, aripiprazole	Asenapine, aripiprazole					Lithium, divalproex, quetiapine	Lurasidone, asenapine, quetiapine XR, aripiprazole, ziprasidone
**Second-line**	Cariprazine, lurasidone	Cariprazine	Olanzapine + litium/divalproex, carbamazepine ER, olanzapine, divalproex	Olanzapine + litium/divalproex, carbamazepine ER, olanzapine, divalproex			Olanzapine + MS	Olanzapine + MS	Cariprazine, ziprasidone, lurasidone	Lamotrigine, divalproex, lithium, cariprazine, olanzapine; MS + SGA; lithium + divalproex; lithium/divalproex + lamotrigine; olanzapine + fluoxetine
**Third-line**	Olanzapine, olanzapine + fluoxetine, quetiapina, divalproex, lamotrigine, ziprasidone, ECT	Olanzapine, quetiapina, divalproex, ziprasidone, ECT	Ziprasidone, divalproex + carbamazepine ER, cariprazine, Lithium + divalproex, ECT		Ziprasidone (combination)	Ziprasidone (combination)	Aripiprazole, arbamazepine, olanzapine	Aripirazole, carbamazepine, olanzapine, paliperidone, risperidone, valproate	Carbamazepine, Olanzapine	Carbamazepine, lithium + carbamazepine/pramipexole, ECT, MS/SGA + bupropion, MS/SGA + SSRI/MAOI
**Fourth-line**				Lurasidone	Carbamazepine, lurasidone, olanzapine, ECT (combination)	Lurasidone, olanzapine, ECT (combination)	Asenapine, olanzapine–fluoxetine, valproate, ziprasidone	Asenapine, olanzapine–fluoxetine, ziprasidone		
**Insufficient evidence**	Aripirazole, asenapine, carbamazepine, Lithium, rTMS	Olanzapine + fluoxetine, lamotrigine, aripirazole, asenapine, carbamazepine, lithium, rTMS, lurasidone	Lithium, lurasidone, quetiapine, paliperidone, risperidone, risperidone + lithium/divalproex, rTMS	Lithium, quetiapine, paliperidone, risperidone, risperidone + lithium/divalproex, rTMS				Paliperidone, haloperidole+ MS, risperidone + MS	Lithium, quetiapine, haloperidol + MS, risperidone + MS	

Legend: *Guidelines:* CANMAT—ISBD: Canadian Network for Mood and Anxiety Treatments—International Society for Bipolar Disorders; WFSBP: World Federation of Societies of Biological Psychiatry; CINP: International College of Neuropsychopharmacology; RANZGP: Royal Australian and New Zealand College of Psychiatrists; *Abbreviations:* D: depressive symptoms during acute mixed depression: ECT: electroconvulsive therapy; M: manic symptoms during acute mixed depression; MS: mood stabilizer; rTMS: repetitive transcranial magnetic stimulation; SGA: second-generation antipsychotic; SSRI: selective-serotonin reuptake inhibitor; MAOI: monoamine oxidase inhibitors; XR: extended release. * In the RANZGP guidelines, there are no levels of recommendations, but a flowchart of choices to follow according to the type of mixed episode and the prevalence of specific symptoms in each episode is provided.

## Data Availability

Not applicable.

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
