# Peer review of "Mixed Depression: A Mini-Review to Guide Clinical Practice and Future Research Developments"

_brainsci, 2022, doi:10.3390/brainsci12010092_

Round 1

Reviewer 1 Report

Choose a tense in the abstract and stick with it-- recommend present tense.

Sentences from lines 91-95 were difficult to understand; unclear.  Also, I do not see where BD-II is explicitly defined.

Line 123:  'originate'

Lines 146-148:  This sentence is unclear.

Lines 246-247:  singular/plural forms used in same sentence for SGA's; confusing

A nice paper with good historical review and state of the current nomenclature, with some attention to current controversies and future directions; table comparing guidelines is very useful for clinicians. The historical summary of how bipolar illness has been conceived by clinicians across the centuries was interesting and useful in tracing the development of thought.  The authors also hint toward future refinement of nosographic criteria, and how that might Influence clinical accuracy and treatment choices.

Author Response

Q1.Choose a tense in the abstract and stick with it-- recommend present tense.

R1. Thank you for the suggestion. We have now corrected the abstract maintaining the present tense.

Q2. Sentences from lines 91-95 were difficult to understand; unclear.  Also, I do not see where BD-II is explicitly defined.

R2. We have now corrected the sentence and added a brief explanation of the difference between BD-I and BD-II.

“Therefore, a mixed episode could only be present in case of a BD-I diagnosis, and not in MDD or BD-II. In fact, manic episodes are pathognomonic of BD-I but not of BD-II, which is instead characterized by an alternation of hypomanic and depressive episodes.”

Q3. Line 123:  'originate'

R3. Thank you very much. We have now corrected the word.

Q4. Lines 146-148:  This sentence is unclear.

R4. We have corrected the sentence as follows:

The cut-off for a diagnosis of MxD is the presence of three symptoms of hypomania or a score at the Hypomania Interview Guide (HIG) >= 8 during a depressive episode.”

Q5. Lines 246-247:  singular/plural forms used in same sentence for SGA's; confusing

R5. Apologize for this mistake. We have now adjusted the sentence to make it clearer.

Q6. A nice paper with good historical review and state of the current nomenclature, with some attention to current controversies and future directions; table comparing guidelines is very useful for clinicians. The historical summary of how bipolar illness has been conceived by clinicians across the centuries was interesting and useful in tracing the development of thought.  The authors also hint toward future refinement of nosographic criteria, and how that might Influence clinical accuracy and treatment choices.

R6. Thank you very much for carefully reading our mansucript and for your insightful suggestions. We hope that, after the revisions. the clarity and the completeness of our paper have improved.

Reviewer 2 Report

Thank you for the pleasure of reviewing this paper which I enjoyed.

The paper addresses mixed depression and hence is more aptly titled so rather than mixed states.

This paper provides a welcome review on mixed depression (or depression with mixed symptoms specifier] especially considering recent clinical guidelines to address treatment in mixed states, at this time of great flux in its nosology. DSM V criteria are currently tentative and up for review. Of necessity, the guidelines do not focus on critique of the literature per se, and this paper better serves that function.

The authors take us through the historical conceptualizations of mixed states, the various iterations of DSM, clinically useful rating scales to aid in the phenomenology and varied versions of  diagnosis, and a helpful table on treatment recommendations from recent guidelines that address mixed states specifically.  It ends with suggestions for better phenotyping and longitudinal work.  It does  a particularly good job of emphasizing that mixed states in and of themselves are concerning,  with worsened prognosis and increased suicidality, which creates a different emphasis than  DSM-V which focuses on mixed states predicting future bipolar dichotomy.

The table outlining recent treatment guidelines and indeed their remarkable heterogeneity, would serve the clinician even better if it had a summary of what definition of mixed states each  guidelines were addressing.  A comment on ICD 11 would also be helpful, perhaps also mentioning its confusing terminology of ‘mixed mood’ for defining mixed anxiety/depression, which must be confusing to clinicians.

Similarly, when commenting on prevalence, the actual definition of mixed state used is very relevant.  Two references are provided for this (31 and 29) which could be updated and broadened even with out spending time on the differing definitions, for the purposes of space.  One possible reference is McIntyre R.S. JD 2015, another, Miller 2016,  but there are others.

While comments enumerate different symptoms used for definition of mixed depression, the subtle reference in section 3 to the different views of how the symptoms present , with symptoms of each pole being constantly present (DSM) versus fluctuating/rapidly alternating symptoms more widely defined in mixed state, could be emphasized as it is currently a challenging diagnostic dilemma for clinicians when the symptoms do not seem concurrently present all the time.

In the section that elucidates causative or worsening factors for mixed states, as well in the discussion of treatment, I would recommend that reference is made to iatrogenic cause or worsening of mixed states, and the careful tapering of antidepressants in particular, to ameliorate or occasionally eliminate mixed states, at least to a simpler depressive picture easier to treat.  The CANMAT/ISBD guidelines of 2020 speak to this [reference 53] but there are also other references in the literature.  It might be helpful to know if other guidelines specifically address this.

In the references to rating scales, most papers include the Altman self -rating scale and the BDRS. 

Better emphasis could be made on clarifying that a depression scale is required in combination with mania rating scales for mixed states where the scale is not constructed for mixed state specifically .  A clarification of which scales are freely available for the clinician would be helpful.

Sach’s combined scale, the clinical monitoring form,  is an example of an integrated scale easily available to the clinician online and could be used in higher risk cases such as those with major depression, family history of bipolar, a strong external validator.

Reference to use of family history and the assessment of mixed symptoms in youth, where they are very prevalent, is also worth mentioning in this review for comprehensiveness with reference to more detailed papers such as the GLAD-PC guidelines and a recent paper on depression preceding the diagnosis of bipolar disorder.

Finally, the conclusion could be stronger in summarizing our challenges and recommendations for the clinician currently and research in the future.

Lines that could benefit from more clarity:

91–94; there are also would seem to be an important typo here on “in case of a bipolar II diagnosis and not in MDD or bipolar II

103-104; ‘criticality in the responsiveness to treatment’.  I think this refers to the important point that mixed states are of concern inherently and not just in terms of their predictive value of bipolar depression.  It is not clear.

113; some authors contrast… Does this mean ‘disagree’?

179- clarity on whether substance abuse is more prevalent in mixed state or if this is a general statement about mood disorders

218- the meaning of “insidious” here is unclear

Table 1:

The heading does not match the contents referring to atypicals rather than all treatments

Author Response

Q1. Thank you for the pleasure of reviewing this paper which I enjoyed.

R1. Thank you very much for carefully reading our manuscript and for your insightful suggestions. We have done our best to address the comments raised and hope that the manuscript has now improved.

Q2. The paper addresses mixed depression and hence is more aptly titled so rather than mixed states.

R2. Thank you for your comment. We have now removed “mixed states” from the title.

Q3. This paper provides a welcome review on mixed depression (or depression with mixed symptoms specifier] especially considering recent clinical guidelines to address treatment in mixed states, at this time of great flux in its nosology. DSM V criteria are currently tentative and up for review. Of necessity, the guidelines do not focus on critique of the literature per se, and this paper better serves that function. The authors take us through the historical conceptualizations of mixed states, the various iterations of DSM, clinically useful rating scales to aid in the phenomenology and varied versions of  diagnosis, and a helpful table on treatment recommendations from recent guidelines that address mixed states specifically.  It ends with suggestions for better phenotyping and longitudinal work.  It does  a particularly good job of emphasizing that mixed states in and of themselves are concerning,  with worsened prognosis and increased suicidality, which creates a different emphasis than  DSM-V which focuses on mixed states predicting future bipolar dichotomy.

R3. We are really grateful for the dedication shown in reviewing our manuscript and really appreciated your comments.

Q4. The table outlining recent treatment guidelines and indeed their remarkable heterogeneity, would serve the clinician even better if it had a summary of what definition of mixed states each  guidelines were addressing.

R4. Thank you. We have provided the definition adopted in each guideline in a separate file which has been submitted a Supplementary Material (Table S1).

Q5. A comment on ICD 11 would also be helpful, perhaps also mentioning its confusing terminology of ‘mixed mood’ for defining mixed anxiety/depression, which must be confusing to clinicians.

R5. Thank you very much for this very important suggestion. We have now added a brief paragraph about ICD-11 in section 4, as reported below

Differently from the DSM-5, the 11th edition of the International Classification of Diseases (ICD-11), classifies MS as a separate diagnostic category. MS are defined as “the presence of several prominent manic and depressive symptoms consistent with those observed in manic episodes and depressive episodes, which either occur simultaneously or alternate very rapidly…” (ICD-11, 2019). It is specified that, when depressive symptoms predominate in a mixed episode, the most common contropolar symptoms are irritability, distractibility, increased verbal production, and psychomotor agitation (Maj M., 2013). Therefore, unlike the DSM-5, the choice of maintaining a specific diagnostic category and including the overlapping symptoms may increase diagnostic sensitivity and ensure more targeted therapeutic strategies. Clinicians should also be aware that the ICD-11 has introduced the diagnostic category "mixed depressive and anxiety disorder", which may represent a potential confounding factor. This category is defined as the co-presence of depressive and anxiety symptoms, not severe enough to warrant a diagnosis of depressive disorder or anxiety disorder (ICD-11, 2019). Therefore, this diagnostic category does not have anything to deal with MS or MxD. Some authors criticize the choice of both current nosographic models to propose a definition of MS that involves a combinatorial approach of symptoms of both polarities rather than specific profiles of different mixed episode subtypes. Indeed, the combinatorial approach would not be able to discriminate among the more serious clinical forms of MxD (Perugi, 2021).

Q6. Similarly, when commenting on prevalence, the actual definition of mixed state used is very relevant.  Two references are provided for this (31 and 29) which could be updated and broadened even with out spending time on the differing definitions, for the purposes of space.  One possible reference is McIntyre R.S. JD 2015, another, Miller 2016,  but there are others.

R6. Thank you for the suggestion. We have added a comment of the difference prevalence rates in Section 5, referring to the paper by Miller et al. 2016.

“The most relevant aspect is related to the set of diagnostic criteria proposed for the definition of MxD, which according to some authors would be too narrow to include the entire spectrum of symptoms. One criticism is the absence of irritable mood and psychomotor agitation in the current DSM-5 diagnostic category of “depressive episode with mixed features”. This is important because prevalence rates may vary depending on the definition of MxD used. For instance, a study by Miller et al. (2016) found that the use of narrower criteria (i.e., DSM-5) allowed to identify 2.6% to 10.8% of MxD in a sample of patients with BD. Conversely, the 14,9% could be diagnosed with MxD using broader criteria (i.e., the presence of subthreshold hypomania concurrent with at least mild depression) (Miller, 2016)”

Q7. While comments enumerate different symptoms used for definition of mixed depression, the subtle reference in section 3 to the different views of how the symptoms present , with symptoms of each pole being constantly present (DSM) versus fluctuating/rapidly alternating symptoms more widely defined in mixed state, could be emphasized as it is currently a challenging diagnostic dilemma for clinicians when the symptoms do not seem concurrently present all the time.

R7. This is a very important aspect indeed. We have emphasized this notion in Section 4, as reported below:

“Another consideration must be made with regard to the choice of including in the definition of MxD also the rapid alternation of the two polarities. This makes it difficult to discriminate between a mixed episode and ultra-fast cycling (mood changing states of polarity over weeks or days) and in particular with ultradian cycling (mood changes occurring within a day) (Maj, M. (2012).”

Q8. In the section that elucidates causative or worsening factors for mixed states, as well in the discussion of treatment, I would recommend that reference is made to iatrogenic cause or worsening of mixed states, and the careful tapering of antidepressants in particular, to ameliorate or occasionally eliminate mixed states, at least to a simpler depressive picture easier to treat.  The CANMAT/ISBD guidelines of 2020 speak to this [reference 53] but there are also other references in the literature.  It might be helpful to know if other guidelines specifically address this.

R8. We agree with the importance of underlying the iatrogenic cause of mixed states. Therefore, we have added a few sentences in this regard in Section 6, as reported below:

“Finally, taking antidepressants during a depressive episode may represent an important cause of development of mixed symptoms (i.e., iatrogenic cause). In this case, main guidelines recommend a careful reduction of the dosage or even a suspension of the antidepressant medication (CANMAT).”

Q9. In the references to rating scales, most papers include the Altman self -rating scale and the BDRS. 

R9. Thank you for suggesting these important scales. We have added them in Section 7.

Q10. Better emphasis could be made on clarifying that a depression scale is required in combination with mania rating scales for mixed states where the scale is not constructed for mixed state specifically.  A clarification of which scales are freely available for the clinician would be helpful.

R10. This part has been clarified as per your suggestion.

“Other scales, such as the MADRS, HAM-D etc… can be used to assess the presence and severity of depressive symptoms. The combination of scales for depression and mania can be useful for clinicians to assess the presence of a MS even if not specifically designed for this purpose.”

Q11. Sach’s combined scale, the clinical monitoring form,  is an example of an integrated scale easily available to the clinician online and could be used in higher risk cases such as those with major depression, family history of bipolar, a strong external validator.

R11. Thank you for making us aware of this useful tool. We have added a paragraph about Sachs’ scale in Section 7.

“Finally,the Clinical Monitoring Form (CMF) proposed by Sachs et al may represent a useful freely available tool in high-risk cases, such as patients with depressive symptoms and a family history of BD (Sachs, 2002)”

Q12. Reference to use of family history and the assessment of mixed symptoms in youth, where they are very prevalent, is also worth mentioning in this review for comprehensiveness with reference to more detailed papers such as the GLAD-PC guidelines and a recent paper on depression preceding the diagnosis of bipolar disorder.

R12. We have added a paragraph in Section 6 about such relevant factors for mixed symptoms, including a mention of the important of assessing these symptoms also in the pediatric population. We have integrated the text using the suggested references.

“The pediatric population represent a poorly studied category which however seems at risk for the development of mixed symptoms. In fact, adolescents with MxD appear to have higher levels of disability, increased severity, and more comorbidities than adolescents with pure MDD or BD (Frazier EA, 2017). During depression, even the family history of BD or completed suicide could be indicative of the possible development of mixed symptoms and manic switch (O'Donovan, 2020)”

Q13. Finally, the conclusion could be stronger in summarizing our challenges and recommendations for the clinician currently and research in the future.

R13. Thank you. We have extended the conclusions further strengthening the important of a clinical characterization and treatment, as follows:

“Our study aimed to provide a synthesis about MxD and guide clinicians’ choices. The present review has shown that current evidence is still unclear in several points. This uncertainty may be related to the lack of neurobiological and epidemiological studies based on recognized diagnostic criteria (Sani et al, 2020). A better phenotyping of patients in clinical practice could solve many questions about the diagnostic orientation (Maj et al, 2020). In this regard, the creation of clinical and research groups on MxD would favor the formulation of more sensitive and specific criteria for the identification of this condition. This may consequently lead to a personalization of care and treatment. In fact, the use of inappropriate medications, such as antidepressants, may worsen the clinical picture instead of ameliorating sympthoms. Of note, longitudinal studies are warranted to evaluate whether the presence of mixed features in individual with MDD may represent a warning for a potential evolution to BD. The present review is important to remember to clinicians that MxD is very common in clinical practice and often associated with a worse outcome and high suicide risk. Future research should focus on these aspects in order to provide clear answers.”

Q14. Lines that could benefit from more clarity: 91–94; there are also would seem to be an important typo here on “in case of a bipolar II diagnosis and not in MDD or bipolar II

R14. Thank you very much for noticing this and apologize for the mistake. We corrected the sentence.

Q15. 103-104; ‘criticality in the responsiveness to treatment’.  I think this refers to the important point that mixed states are of concern inherently and not just in terms of their predictive value of bipolar depression.  It is not clear.

R15. Thank you for this suggestion. We have now corrected the sentence as follows:

“The precise identification of a depressive episode with mixed characteristics is of great importance for the subsequent diagnostic framework and treatment planning, because of the significantly worse course and the peculiar responsiveness to treatment.”

Q16. 113; some authors contrast… Does this mean ‘disagree’?

R16. Yes, thank you, we changed the sentence as follows:

“Some authors disagree with the conceptualization…”

Q17. 179- clarity on whether substance abuse is more prevalent in mixed state or if this is a general statement about mood disorders

R17. Thank you. Indeed the reference used was not correct and there is no substantial evidence in literature to justify this idea. Therefore, we have decided to delete the sentence.

Q18. 218- the meaning of “insidious” here is unclear

R18. We have deleted the term insidious for better clarity.

Q19. Table 1: The heading does not match the contents referring to atypicals rather than all treatments.

R19. Thank you for noticing the mistake. We have changed the heading as follows…

“Pharmacological treatment for the acute phase of mixed depression”